# Efficient generative modeling of protein sequences using simple autoregressive models

Jeanne Trinquier[1,2], Guido Uguzzoni[3,4], Andrea Pagnani[3,4,5], Francesco Zamponi[2] & Martin Weigt [1✉]

Generative models emerge as promising candidates for novel sequence-data driven approaches to protein design, and for the extraction of structural and functional information about proteins deeply hidden in rapidly growing sequence databases. Here we propose simple autoregressive models as highly accurate but computationally efficient generative sequence models. We show that they perform similarly to existing approaches based on Boltzmann machines or deep generative models, but at a substantially lower computational cost (by a factor between $10^2$ and $10^3$). Furthermore, the simple structure of our models has distinctive mathematical advantages, which translate into an improved applicability in sequence generation and evaluation. Within these models, we can easily estimate both the probability of a given sequence, and, using the model's entropy, the size of the functional sequence space related to a specific protein family. In the example of response regulators, we find a huge number of ca. $10^{68}$ possible sequences, which nevertheless constitute only the astronomically small fraction $10^{-80}$ of all amino-acid sequences of the same length. These findings illustrate the potential and the difficulty in exploring sequence space via generative sequence models.

[1] Sorbonne Université, CNRS, Institut de Biologie Paris Seine, Biologie Computationnelle et Quantitative LCQB, F-75005 Paris, France. [2] Laboratoire de Physique de l'Ecole Normale Supérieure, ENS, Université PSL, CNRS, Sorbonne Université, Université de Paris, F-75005 Paris, France. [3] Department of Applied Science and Technology (DISAT), Politecnico di Torino, Corso Duca degli Abruzzi 24, I-10129 Torino, Italy. [4] Italian Institute for Genomic Medicine, IRCCS Candiolo, SP-142, I-10060 Candiolo (TO), Italy. [5] INFN Sezione di Torino, Via P. Giuria 1, I-10125 Torino, Italy. ✉email: martin.weigt@sorbonne-universite.fr

The impressive growth of sequence databases is prompted by increasingly powerful techniques in data-driven modeling, helping to extract the rich information hidden in raw data. In the context of protein sequences, unsupervised learning techniques are of particular interest: only about 0.25% of the more than 200 million amino-acid sequences currently available in the Uniprot database[1] have manual annotations, which can be used for supervised methods.

Unsupervised methods may benefit from evolutionary relationships between proteins: while mutations modify amino-acid sequences, selection keeps their biological functions and their three-dimensional structures remarkably conserved. The Pfam protein family database[2], e.g., lists more than 19,000 families of homologous proteins, offering rich datasets of sequence-diversified but functionally conserved proteins.

In this context, generative statistical models are rapidly gaining interest. The natural sequence variability across a protein family is captured via a probability $P(a_1,..., a_L)$ defined for all amino-acid sequences $(a_1,..., a_L)$. Sampling from $P(a_1,..., a_L)$ can be used to generate new, non-natural amino-acid sequences, which is an ideal case that should be statistically indistinguishable from the natural sequences. However, the task of learning $P(a_1,..., a_L)$ is highly non-trivial: the model has to assign probabilities to all $20^L$ possible amino-acid sequences. For typical proteins of lengths, $L = 50 - 500$, this accounts for $10^{65} - 10^{650}$ values, to be learned from the $M = 10^3 - 10^6$ sequences contained in most protein families. Selecting adequate generative model architectures is thus of outstanding importance.

The currently best explored generative models for proteins are so-called coevolutionary models[3], such as those constructed by the Direct Coupling Analysis (DCA)[4–6] (a more detailed review of the state of the art is provided below). They explicitly model the usage of amino acids in single positions (i.e., residue conservation) and correlations between pairs of positions (i.e., residue coevolution). The resulting models are mathematically equivalent to Potts models[7] in statistical physics, or to Boltzmann machines in statistical learning[8]. They have found numerous applications in protein biology.

The effect of amino-acid mutations is predicted via the log-ratio log $\{P(\text{mutant})/P(\text{wildtype})\}$ between mutant and wildtype probabilities. Strong correlations to mutational effects determined experimentally via deep mutational scanning have been reported[9,10]. Promising applications are the data-driven design of mutant libraries for protein optimization[11–13], and the use of Potts models as sequence landscapes in quantitative models of protein evolution[14,15].

Contacts between residues in the protein fold are extracted from the strongest epistatic couplings between double mutations, i.e., from the direct couplings giving the name to DCA[6]. These couplings are essential input features in the wave of deep-learning (DL) methods, which currently revolutionize the field of protein-structure prediction[16–19].

The generative implementation bmDCA[5] is able to generate artificial but functional amino-acid sequences[20,21]. Such observations suggest novel but almost unexplored approaches towards data-driven protein design, which complement current approaches based mostly on large-scale experimental screening of randomized sequence libraries or time-intensive bio-molecular simulation, typically followed by sequence optimization using directed evolution, cf. refs. [22,23] for reviews.

Here we propose a simple model architecture called arDCA, based on a shallow (one-layer) autoregressive model paired with generalized logistic regression. Such models are computationally very efficient, they can be learned in few minutes, as compared to days for bmDCA and more involved architectures. Nevertheless, we demonstrate that arDCA provides highly accurate generative models, comparable to the state of the art in mutational-effect and residue-contact prediction. Their simple structure makes them more robust in the case of limited data. Furthermore, and this may have important applications in homology detection[24], our autoregressive models are the only generative models we know about, which allow for calculating exact sequence probabilities, and not only non-normalized sequence weights. Thereby arDCA enables the comparison of the same sequence in different models for different protein families. Last but not least, the entropy of arDCA models, which is related to the size of the functional sequence space associated with a given protein family, can be computed much more efficiently than in bmDCA.

Before proceeding, we provide here a short review of the state of the art in generative protein modeling. The literature is extensive and rapidly growing, so we will concentrate on the methods being most directly relevant as compared to the scope of our work.

We focus on generative models purely based on sequence data. The sequences belonging to homologous protein families and are given in form of multiple sequence alignments (MSA), i.e., as a rectangular matrix $\mathcal{D} = (a_i^m | i = 1, ..., L; m = 1, ..., M)$ containing $M$ aligned proteins of length $L$. The entries $a_i^m$ equal either one of the standard 20 amino acids or the alignment gap "–". In total, we have $q = 21$ possible different symbols in $\mathcal{D}$. The aim of unsupervised generative modeling is to learn a statistical model $P(a_1,..., a_L)$ of (aligned) full-length sequences, which faithfully reflects the variability found in $\mathcal{D}$: sequences belonging to the protein family of interest should have comparably high probabilities, unrelated sequences very small probabilities. Furthermore, a new artificial MSA $\mathcal{D}'$ sampled sequence by sequence from model $P(a_1,..., a_L)$ should be statistically and functionally indistinguishable from the natural aligned MSA $\mathcal{D}$ given as input.

A way to achieve this goal is the above-mentioned use of Boltzmann-machine learning based on conservation and coevolution, which leads to pairwise-interacting Potts models, i.e., bmDCA[5], and related methods[25–27]. An alternative implementation of bmDCA, including the decimation of statistically irrelevant couplings, has been presented in[28] and is the one used as a benchmark in this work; the Mi3 package[29] also provides a GPU-based accelerated implementation.

However, Potts models or Boltzmann machines are not the only generative-model architectures explored for protein sequences. Latent-variable models like Restricted Boltzmann machines[30] or Hopfield-Potts models[31] learn dimensionally reduced representations of proteins; using sequence motifs, they are able to capture groups of collectively evolving residues[32] better than DCA models, but are less accurate in extracting structural information from the learning MSA[31].

An important class of generative models based on latent variables are variational autoencoders (VAE), which achieve dimensional reduction, but in the flexible and powerful set of deep learning. The DeepSequence implementation[33] was originally designed and tested for predicting the effects of mutations around a given wild type. It currently provides one of the best mutational-effect predictors, and we will show below that arDCA provides comparable quality of prediction for this specific task. The DeepSequence code has been modified in[34] to explore its capacities in generating artificial sequences being statistically indistinguishable from the natural MSA; it was shown that its performance was substantially less accurate than bmDCA. Another implementation of a VAE was reported in[35]; also in this case the generative performances are inferior to bmDCA, but the organization of latent variables was shown to carry significant information on functionality. Furthermore, some generated mutant sequences were successfully tested experimentally. Interestingly, it was also shown that learning VAE on unaligned

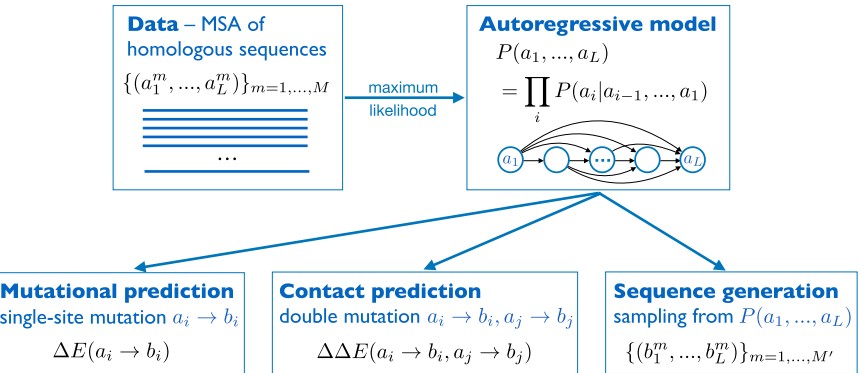

**Fig. 1 Schematic representation of the arDCA approach: Starting from an MSA of homologous sequences, we use maximum-likelihood inference to learn an autoregressive model, which factorizes the joint sequence probability $P(a_1, ..., a_L)$ into conditional single-residue probabilities $P(a_i|a_{i-1},..., a_1)$.** Defining the statistical energy $E(a_1, ..., a_L) = -\log P(a_1, ..., a_L)$ of a sequence, we consequently predict mutational effects and contacts as statistical energy changes when substituting residues individually or in pairs, and we design new sequences by sampling from $P(a_1, ..., a_L)$.

sequences decreases the performance as compared to pre-aligned MSA as used by all before-mentioned models. This observation was complemented by Ref. [36], which reported a VAE implementation trained on non-aligned sequences from UniProt, with length $10 < L < 1000$. The VAE had good reconstruction accuracy for small $L < 200$, which however dropped significantly for larger $L$. The latent space also in this case shows an interesting organization in terms of function, which was used to generate *in silico* proteins with desired properties, but no experimental test was provided. The paper does not report any statistical test of the generative properties (such as a Pearson correlation of two-point correlations), and the publicly not yet available code makes a quantitative comparison to our results currently impossible.

Another interesting DL architecture is that of a Generative Adversarial Network (GAN), which was explored in [37] on a single family of aligned homologous sequences. While the model has a very large number of trainable parameters (~60 M), it seems to reproduce well the statistics of the training MSA, and most importantly, the authors could generate an enzyme with only 66% identity to the closest natural one, which was still found to be functional in vitro. An alternative implementation of the same architecture was presented in [38], and applied to the design of antibodies; also in this case the resulting sequences were validated experimentally.

Not all generative models for proteins are based on sequence ensembles. Several research groups explored the possibility of generating sequences with given three-dimensional structure [39–41], e.g. via a VAE [42] or a Graph Neural Network [43], or by inverting structural prediction models [44−47]. It is important to stress that this is a very different task from ours (our work does not use structure), so it is difficult to perform a direct comparison between our work and these ones. It would be interesting to explore, in future work, the possibility to unify the different approaches and to use sequence and structure jointly for constructing improved generative models.

In summary, for the specific task of interest here, namely, generate an artificial MSA statistically indistinguishable from the natural one, one can take as reference models bmDCA [28,5] in the context of Potts-model-like architectures, and DeepSequence [33] in the context of deep networks. We will show in the following that arDCA performs comparably to bmDCA, and better than DeepSequence, at a strongly reduced computational cost. From anecdotal evidence in the works mentioned above, and in agreement with general observations in machine learning, it appears that deep architectures may be more powerful than shallow architectures, provided that very large datasets and

computational resources are available [33]. Indeed, we will show that for the related task of single-mutation predictions around a wild type, DeepSequence outperforms arDCA on rich datasets, while the inverse is true on small datasets.

## Results

**Autoregressive models for protein families**. Here we propose a computationally efficient approach based on autoregressive models, cf. Fig. 1 for an illustration of the approach and the model architecture. We start from the exact decomposition

$$P(a_1, ..., a_L) = P(a_1) \cdot P(a_2|a_1) \cdots P(a_L|a_{L-1}, ..., a_1), \quad (1)$$

of the joint probability of a full-length sequence into a product of more and more involved conditional probabilities $P(a_i|a_{i-1},..., a_1)$ of the amino acids $a_i$ in single positions, conditioned to all previously seen positions $a_{i-1},..., a_1$. While this decomposition is a direct consequence of Bayes' theorem, it suggests an important change in viewpoint on generative models: while learning the full $P(a_1,..., a_L)$ from the input MSA $\mathcal{D}$ is a task of unsupervised learning (sequences are not labeled), learning the factors $P(a_i|a_{i-1},..., a_1)$ becomes a task of supervised learning, with $(a_{i-1},..., a_1)$ being the input (feature) vector, and $a_i$ the output label (in our case a categorical $q$-state label). We can thus build on the full power of supervised learning, which is methodologically more explored than unsupervised learning [48–50].

In this work, we choose the following parameterization, previously used in the context of statistical mechanics of classical [51] and quantum [52] systems:

$$P(a_i|a_{i-1}, ..., a_1) = \frac{\exp\left\{h_i(a_i) + \sum_{j=1}^{i-1} J_{ij}(a_i, a_j)\right\}}{z_i(a_{i-1}, ..., a_1)}, \quad (2)$$

with $z_i(a_{i-1}, ..., a_1) = \sum_{a_i} \exp\{h_i(a_i) + \sum_{j=1}^{i-1} J_{ij}(a_i, a_j)\}$ being a normalization factor. In machine learning, this parameterization is known as soft-max regression, the generalization of logistic regression to multi-class labels [50]. This choice, as detailed in the section "Methods", enables a particularly efficient parameter learning by likelihood maximization, and leads to a speedup of 2−3 orders of magnitude over bmDCA, as is reported in Table 1. Because the resulting model is parameterized by a set of fields $h_i(a)$ and couplings $J_{ij}(a, b)$ as in DCA, we dub our method as arDCA.

Besides comparing the performance of this model to bmDCA and DeepSequence, we will also use simple "fields-only" models, also known as profile models or independent-site models. In these models, the joint probability of all positions in a sequence

**Table 1 The table summarizes the data used (protein families, sequence lengths $L$ and numbers $M$, together with the Pearson correlations between empirical and model-generated connected correlations $C_{ij}$ and $C_{ijk}$ for bmDCA, for arDCA using entropic or direct positional orders, and for DeepSequence (the highest values are emphasized in bold).**

| | $L$ | $M$ | $C_{ij}$ ent. arDCA | $C_{ij}$ dir. arDCA | $C_{ij}$ bmDCA | $C_{ij}$ DeepSeq | $C_{ijk}$ ent. arDCA | $C_{ijk}$ dir. arDCA | $C_{ijk}$ bmDCA | $C_{ijk}$ DeepSeq | entropy arDCA | entropy bmDCA | t/min arDCA | t/min bmDCA |
|---|---|---|---|---|---|---|---|---|---|---|---|---|---|---|
| PF00014 | 53 | 13600 | **0.97** | 0.96 | 0.95 | 0.81 | **0.84** | 0.82 | 0.83 | 0.80 | 1.2 | 1.5 | **1** | 204 |
| PF00076 | 70 | 137605 | **0.97** | **0.97** | **0.97** | 0.84 | 0.78 | 0.76 | **0.85** | 0.77 | 1.6 | 1.7 | 19 | 2088 |
| PF00595 | 80 | 36690 | 0.96 | 0.95 | **0.97** | 0.93 | 0.87 | 0.87 | **0.92** | 0.65 | 1.2 | 1.5 | 8 | 4003 |
| PF00072 | 112 | 823798 | 0.96 | **0.96** | 0.93 | 0.95 | 0.89 | 0.88 | 0.88 | **0.92** | 1.4 | 1.8 | 9 | 1489 |
| PF13354 | 202 | 7515 | **0.97** | 0.96 | 0.95 | 0.95 | **0.93** | 0.91 | 0.92 | 0.92 | 0.9 | 1.2 | **10** | 3905 |

The entropies/site and computational running times for model learning (on a single Intel Xeon E5-2620 v4 2.10 GHz CPU) is also provided for arDCA and bmDCA. Best values for each measure are evidenced. Similar results for the 32 protein families with deep-mutational scanning data are given in the Supplementary Table 2.

factorizes overall positions, $P(a_1,..., a_L) = \prod_{i=1,...,L} f_i(a_i)$, without any conditioning to the sequence context. Using maximum-likelihood inference, each factor $f_i(a_i)$ equals the empirical frequency of amino acid $a_i$ in column $i$ of the input MSA $\mathcal{D}$.

A few remarks are needed.

Eq. (2) has striking similarities to standard DCA[4], but also important differences. The two have exactly the same number of parameters, but their meaning is quite different. While DCA has symmetric couplings $J_{ij}(a, b) = J_{ji}(b, a)$, the parameters in Eq. (2) are directed and describe the influence of site $j$ on site $i$ for $j < i$ only, i.e., only one triangular part of the $J$-matrix is filled.

The inference in arDCA is very similar to plmDCA[53], i.e., to DCA based on pseudo-likelihood maximization[54]. In particular, both in arDCA and plmDCA the gradient of the likelihood can be computed exactly from the data, while in bmDCA it has to be estimated via Monte Carlo Markov Chain (MCMC), which requires the introduction of additional hyperparameters (such as the number of chains, the mixing time, etc.) that can have an important impact on the quality of the inference, see[55] for a recent detailed study.

In plmDCA each $a_i$ is, however, conditioned to *all* other $a_j$ in the sequence, and not only by partial sequences. The resulting directed couplings are usually symmetrized akin to standard Potts models. On the contrary, the $J_{ij}(a, b)$ that appear in arDCA cannot be interpreted as "direct couplings" in the DCA sense, cf. below for details on arDCA-based contact prediction. However, plmDCA has limited capacities as a generative model[5]: symmetrization moves parameters away from their maximum-likelihood value, probably causing a loss in model accuracy. No such symmetrization is needed for arDCA.

arDCA, contrary to all other DCA methods, allows for calculating the probabilities of single sequences. In bmDCA, we can only determine sequence weights, but the normalizing factor, i.e., the partition function, remains inaccessible for exact calculations; expensive thermodynamic integration via MCMC sampling is needed to estimate it. The conditional probabilities in arDCA are individually normalized; instead of summing over $q^L$ sequences, we need to sum $L$-times over the $q$ states of individual amino acids. This may turn out as a major advantage when the same sequence in different models shall be compared, as in homology detection and protein family assignment[56,57], cf. the example given below.

The ansatz in Eq. (2) can be generalized to more complicated relations. We have tested a two-layer architecture but did not observe advantages over the simple soft-max regression, as will be discussed at the end of the paper.

Thanks, in particular, to the possibility of calculating the gradient exactly, arDCA models can be inferred much more efficiently than bmDCA models. Typical inference times are given in Table 1 for five representative families, and show a speedup of about 2–3 orders of magnitude with respect to the bmDCA implementation of[28], both running on a single Intel Xeon E5-2620 v4 2.10 GHz CPU. We also tested the Mi3 package[29], which is able to learn similar bmDCA models in a time of about 60 min for the PF00014 family and 900 min for the PF00595 family, while running on two TITAN RTX GPUs, thus remaining much more computationally demanding than arDCA.

**The positional order matters**. Eq. (1) is valid for any order of the positions, i.e., for any permutation of the natural positional order in the amino-acid sequences. This is no longer true when we parameterize the $P(a_i|a_{i-1},..., a_1)$ according to Eq. (2). Different orders may give different results. In Supplementary Note 1, we show that the likelihood depends on the order and that we can optimize over orders. We also find that the best orders are

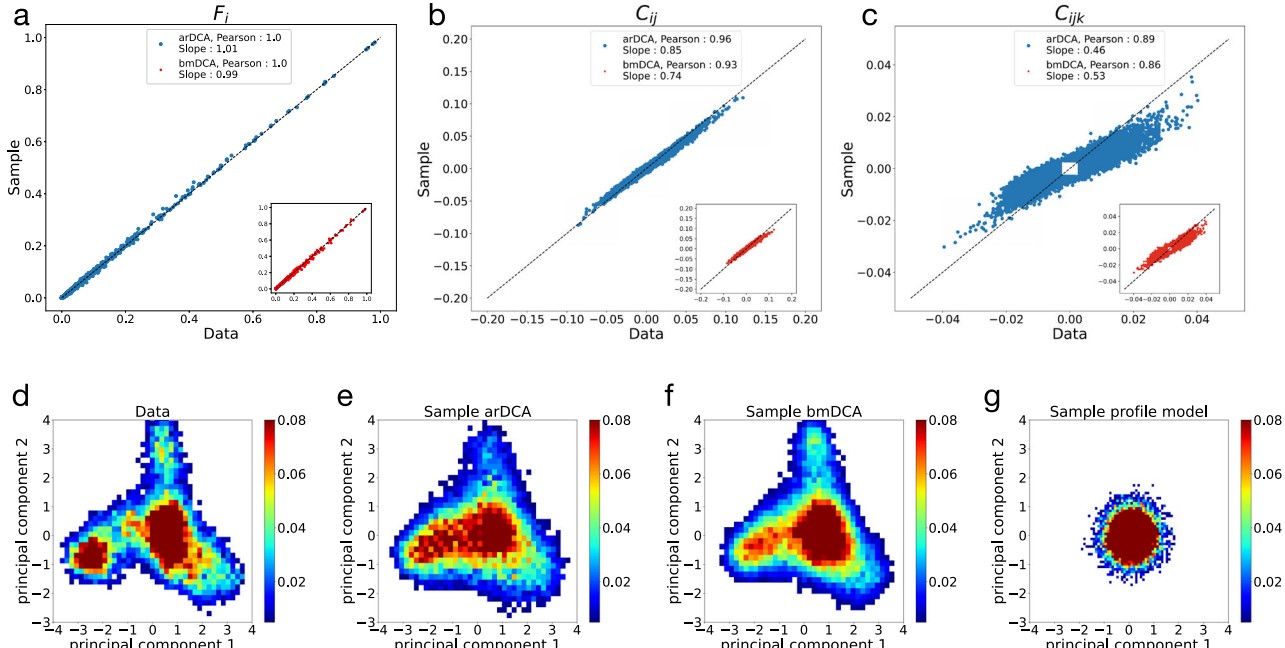

**Fig. 2 Generative properties of arDCA for PF00072.** Panels **a–c** compare the single-site frequencies $f_i(a)$ and two-site and three-site connected correlations $C_{ij}(a, b)$ and $C_{ijk}(a, b, c)$ found in the sequence data and samples from models, for arDCA (blue) and bmDCA (red). Panels **d–g** show different samples projected onto the first two principal components of the natural data. Datasets are the natural MSA (**d**) and samples from arDCA (**e**), bmDCA (**f**), and a profile model (**g**). Results for other protein families are shown in Supplementary Figs. 5–6.

correlated to the entropic order, where we select first the least entropic, *i.e.* most conserved, variables, progressing successively towards the most variable positions of highest entropy. The site entropy $s_i = -\sum_a f_i(a)\log f_i(a)$ can be directly calculated from the empirical amino-acid frequencies $f_i(a)$ of all amino acids $a$ in site $i$.

Because the optimization over the possible $L!$ site orderings is very time-consuming, we use the entropic order as a practical heuristic choice. In all our tests, described in the next sections, the entropic order does not perform significantly worse than the best-optimized order we found.

A close-to-entropic order is also attractive from the point of view of interpretation. The most conserved sites come first. If the amino acid on those sites is the most frequent one, basically no information is transmitted further. If, however, a sub-optimal amino acid is found in a conserved position, this has to be compensated by other mutations, i.e., necessarily by more variable (more entropic) positions. Also, the fact that variable positions come last, and are modeled as depending on all other amino acids, is well interpretable: these positions, even if highly variable, are not necessarily unconstrained, but they can be used to finely tune the sequence to any sub-optimal choices done in earlier positions.

For this reason, all coming tests are done using increasing entropic order, i.e., with sites ordered before model learning by increasing empirical $s_i$ values. Supplementary Figs. 1–3 shows a comparison with alternative orderings, such as the direct one (from 1 to $L$), several random ones, and the optimized one, cf. also Table 1 for some results.

**arDCA provides accurate generative models**. To check the generative property of arDCA, we compare it with bmDCA[5], i.e., the most accurate generative version of DCA obtained via Boltzmann machine learning. bmDCA was previously shown to be generative not only in a statistical sense, but also in a biological one: sequences generated by bmDCA were shown to be

statistically indistinguishable from natural ones, and most importantly, functional in vivo for the case of chorismate mutase enzymes[20]. We also compare the generative property of arDCA with DeepSequence[33,34] as a prominent representative of deep generative models.

To this aim, we compare the statistical properties of natural sequences with those of independently and identically distributed (i.i.d.) samples drawn from the different generative models $P(a_1,..., a_L)$. At this point, another important advantage of arDCA comes into play: while generating i.i.d. samples from, e.g., a Potts model requires MCMC simulations, which in some cases may have very long decorrelation times and thus become tricky and computationally expensive[28,55] (cf. also Supplementary Note 2 and Supplementary Fig. 4), drawing a sequence from the arDCA model $P(a_1,..., a_L)$ is very simple and does not require any additional parameter. The factorized expression Eq. (1) allows for sampling amino acids position by position, following the chosen positional order, cf. the detailed description in Supplementary Note 2.

Figures 2a–c show the comparison of the one-point amino-acid frequencies $f_i(a)$, and the connected two-point and three-point correlations

$$C_{ij}(a, b) = f_{ij}(a, b) - f_i(a)f_j(b) ,$$
$$C_{ijk}(a, b, c) = f_{ijk}(a, b, c) - f_{ij}(a, b)f_k(c) - f_{ik}(a, c)f_j(b) \quad (3)$$
$$- f_{jk}(b, c)f_i(a) + 2f_i(a)f_j(b)f_k(c),$$

of the data with those estimated from a sample of the arDCA model. Results are shown for the response-regulator Pfam family PF00072[2]. Other proteins are shown in Table 1 and Supplementary Note 3, Supplementary Figs. 5–6. We find that, for these observables, the empirical and model averages coincide very well, equally well, or even slightly better than for the bmDCA case. In particular, for the one-point and two-point quantities, this is quite surprising: while bmDCA fits them explicitly, i.e., any

deviation is due to the imperfect fitting of the model, arDCA does not fit them explicitly, and nevertheless obtains higher precision.

In Table 1, we also report the results for sequences sampled from DeepSequence[33]. While its original implementation aims at scoring individual mutations, cf. Section "Predicting mutational effects via in-silico deep mutational scanning", we apply the modification of ref. [34] allowing for sequence sampling. We observe that for most families, the two-point and three-point correlations of the natural data are significantly less well reproduced by DeepSequence than by both DCA implementations, confirming the original findings of ref. [34]. Only in the largest family, PF00072 with more than 800,000 sequences, DeepSequence reaches comparable or, in the case of the three-point correlations, even superior performance.

A second test of the generative property of arDCA is given by Fig. 2d–g. Panel d shows the natural sequences projected onto their first two principal components (PC). The other three panels show generated data projected onto the same two PCs of the natural data. We see that both arDCA and bmDCA reproduce quite well the clustered structure of the response-regulator sequences (both show a slightly broader distribution than the natural data, probably due to the regularized inference of the statistical models). On the contrary, sequences generated by a profile model $P_{\text{prof}}(a_1,..., a_L) = \prod_i f_i(a_i)$ assuming independent sites, do not show any clustered structure: the projections are concentrated around the origin in PC space. This indicates that their variability is almost unrelated to the first two principal components of the natural sequences.

From these observations, we conclude that arDCA provides excellent generative models, of at least the same accuracy as bmDCA. This suggests fascinating perspectives in terms of data-guided statistical sequence design: if sequences generated from bmDCA models are functional, also arDCA-sampled sequences should be functional. But this is obtained at a much lower computational cost, cf. Table 1 and without the need to check for convergence of MCMC, which makes the method scalable to much bigger proteins.

**Predicting mutational effects via in-silico deep mutational scanning.** The probability of a sequence is a measure of its goodness. For high-dimensional probability distributions, it is generally convenient to work with log probabilities. Using inspiration from statistical physics, we introduce a statistical energy

$$E(a_1,...,a_L) = -\log P(a_1,...,a_L),\quad (4)$$

as the negative log probability. We thus expect functional sequences to have very low statistical energies, while unrelated sequences show high energies. In this sense, statistical energy can be seen as a proxy of (negative) fitness. Note that in the case of arDCA, the statistical energy is not a simple sum over the model parameters as in DCA, but contains also the logarithms of the local partition functions $z_i(a_{i-1},..., a_1)$, cf. Eq. (2).

Now, we can easily compare two sequences differing by one or few mutations. For a single mutation $a_i \rightarrow b_i$, where amino acid $a_i$ in position $i$ is substituted with amino acid $b_i$, we can determine the statistical-energy difference

$$\Delta E(a_i \rightarrow b_i) = -\log \frac{P(a_1,...,a_{i-1},b_i,a_{i+1},....,a_L)}{P(a_1,...,a_{i-1},a_i,a_{i+1},....,a_L)}.\quad (5)$$

If negative, the mutant sequence has lower statistical energy; the mutation $a_i \rightarrow b_i$ is thus predicted to be beneficial. On the contrary, a positive $\Delta E$ predicts a deleterious mutation. Note that, even if not explicitly stated on the left-hand side of Eq. (5), the mutational score $\Delta E(a_i \rightarrow b_i)$ depends on the whole sequence

background $(a_1,..., a_{i-1}, a_{i+1},....,a_L)$ it appears in, i.e., on all other amino acids $a_j$ in all positions $j \neq i$.

It is now easy to perform an in-silico deep mutational scan, i.e., to determine all mutational scores $\Delta E(a_i \rightarrow b_i)$ for all positions $i = 1, ..., L$ and all target amino acids $b_i$ relative to some reference sequence. In Fig. 3a, we compare our predictions with experimental data over more than 30 distinct experiments and wildtype proteins, and with state-of-the-art mutational-effect predictors. These contain in particular the predictions using plmDCA (aka evMutation[10]), variational autoencoders (DeepSequence[33]), evolutionary distances between wildtype, and the closest homologs showing the considered mutation (GEMME[58])—all of these methods take, in technically different ways, the context-dependence of mutations into account. We also compare it to the context-independent prediction using the above-mentioned profile models.

It can be seen that the context-dependent predictors outperform systematically the context-independent predictor, in particular for large MSA in prokaryotic and eukaryotic proteins. The four context-dependent models perform in a very similar way. There is a little but systematic disadvantage for plmDCA, which was the first published predictor of the ones considered here.

The situation is different in the typically smaller and less diverged viral protein families. In this case, DeepSequence, which relies on data-intensive deep learning, becomes unstable. It becomes also harder to outperform profile models, e.g., plmDCA does not achieve this. arDCA performs similarly or, in one out of four cases, substantially better than the profile model.

To go into more detail, we have compared more quantitatively the predictions of arDCA and DeepSequence, currently considered as the state-of-the-art mutational predictor. In Fig. 3b, we plot the performance of the two predictors against each other, with the symbol size being proportional to the number of sequences in the training MSA of natural homologs. Almost all dots are close to the diagonal (apart from few viral datasets), with **17**/32 datasets having a better arDCA prediction and **15**/32 giving an advantage to DeepSequence. The figure also shows that arDCA tends to perform better on smaller datasets, while DeepSequence takes over on larger datasets. In Supplementary Fig. 7, we have also measured the correlations between the two predictors. Across all prokaryotic and eukaryotic datasets, the two show high correlations in the range of 82–95%. These values are larger than the correlations between predictions and experimental results, which are in the range of 50–60% for most families. This observation illustrates that both predictors extract a highly similar signal from the original MSA, but this signal may be quite different from the experimentally measured phenotype. Many experiments actually provide only rough proxies for protein fitness, like e.g. protein stability or ligand-binding affinity. To what extent such variable underlying phenotypes can be predicted by unsupervised learning based on homologous MSA thus remains an open question.

We thus conclude that arDCA permits a fast and accurate prediction of mutational effects, in line with some of the state-of-the-art predictors. It systematically outperforms profile models and plmDCA and is more stable than DeepSequence in the case of limited datasets. This observation, together with the better computational efficiency of arDCA, suggests that DeepSequence should be used for predicting mutational effects for individual proteins represented by very large homologous MSA, while arDCA is the method of choice for large-scale studies (many proteins) or small families. GEMME, based on phylogenetic information, astonishingly performs very similarly to arDCA, even if the information taken into account seems different.

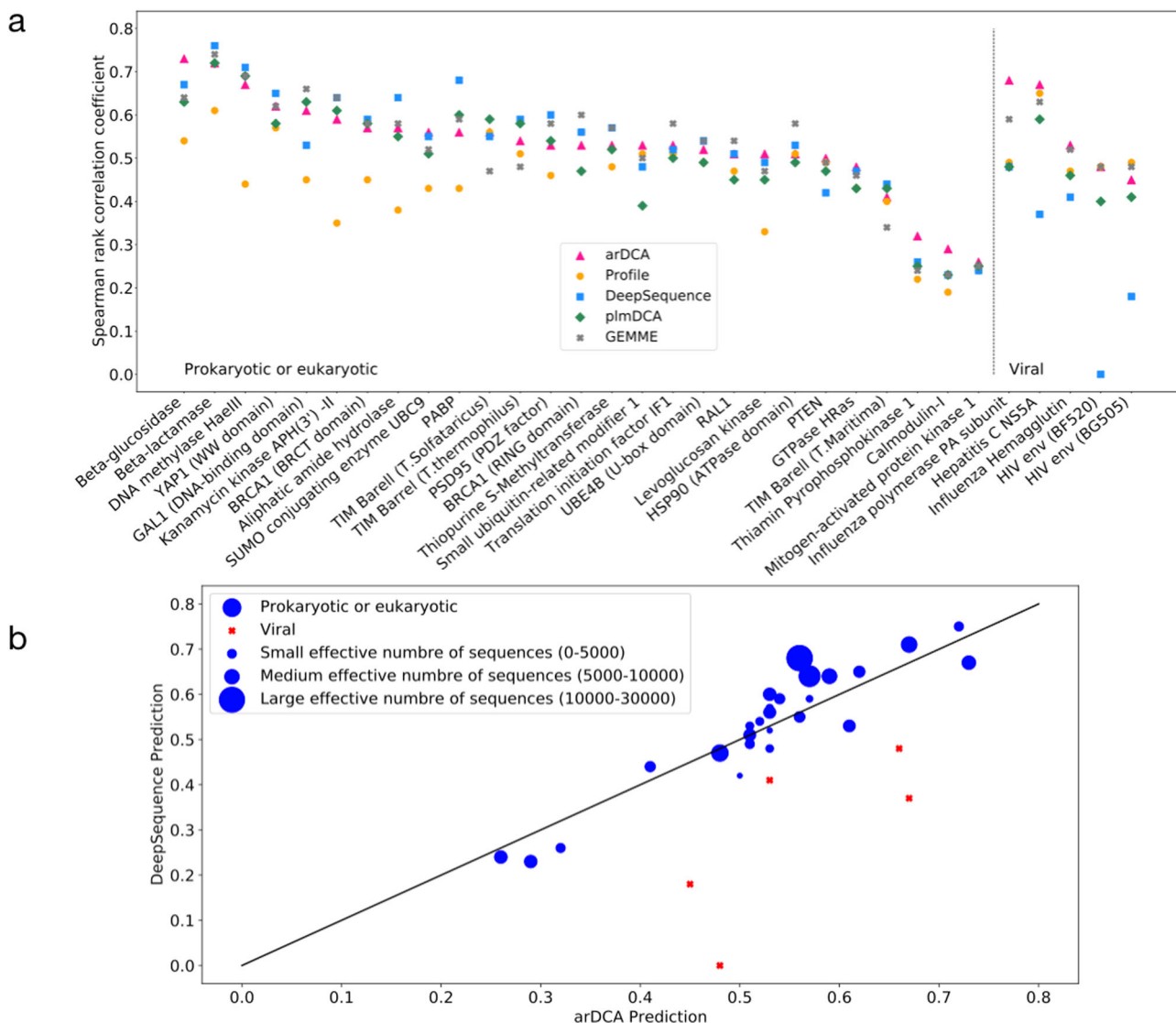

**Fig. 3 Prediction of mutational effects by arDCA.** Panel **a** shows the Spearman rank correlation between results of 32 deep-mutational scanning experiments and various computational predictions. We compare arDCA with profile models, plmDCA (aka evMutation[10]), DeepSequence[33], and GEMME[59], which currently are considered the state of the art. Detailed information about the datasets and the generative properties of arDCA on these datasets are provided in Supplementary Note 4. Panel **b** shows a more detailed comparison between arDCA and DeepSequence, the symbol size is proportional to the sequence number in the training MSA for prokaryotic and eukaryotic datasets (blue dots). Viral datasets are indicated by red squares.

**Extracting epistatic couplings and predicting residue-residue contacts.** The best-known application of DCA is the prediction of residue-residue contacts via the strongest direct couplings[6]. As argued before, the arDCA parameters are not directly interpretable in terms of direct couplings. To predict contacts using arDCA, we need to go back to the biological interpretation of DCA couplings: they represent epistatic couplings between pairs of mutations[59]. For a double mutation $a_i \rightarrow b_i, a_j \rightarrow b_j$, epistasis is defined by comparing the effect of the double mutation with the sum of the effects of the single mutations, when introduced individually into the wildtype background:

$$\Delta\Delta E(b_i, b_j) = \Delta E(a_i \rightarrow b_i, a_j \rightarrow b_j) \\ - \Delta E(a_i \rightarrow b_i) - \Delta E(a_j \rightarrow b_j), \qquad (6)$$

where the $\Delta E$ in arDCA is defined in analogy to Eq. (5). The epistatic effect $\Delta\Delta E(b_i, b_j)$ provides an effective direct coupling between amino acids $b_i, b_j$ in sites $i, j$. In standard DCA, $\Delta\Delta E(b_i, b_j)$ is actually given by the direct coupling $J_{ij}(b_i, b_j) - J_{ij}(b_i, a_j) - J_{ij}(a_i, b_j) + J_{ij}(a_i, a_j)$ between sites $i$ and $j$.

For contact prediction, we can treat these effective couplings in the standard way (compute the Frobenius norm in zero-sum gauge, apply the average product correction, cf. Supplementary Note 5 for details). The results are represented in Fig. 4 (cf. also Supplementary Figs. 8–10). The contact maps predicted by arDCA and bmDCA are very similar, and both capture very well the topological structure of the native contact map. The arDCA method gives in this case a few more false positives, resulting in a slightly lower positive predictive value (panel c). However, note that the majority of the false positives for both predictors are concentrated in the upper right corner of the contact maps, in a region where the largest subfamily of response-regulators domains, characterized by the coexistence with a Trans_reg_C DNA-binding domain (PF00486) in the same protein, has a homo-dimerization interface.

One difference should be noted: for arDCA, the definition of effective couplings via epistatic effects depends on the reference sequence $(a_1, ..., a_L)$, in which the mutations are introduced; this is not the case in DCA. So, in principle, each sequence might give a different contact prediction, and accurate contact prediction in

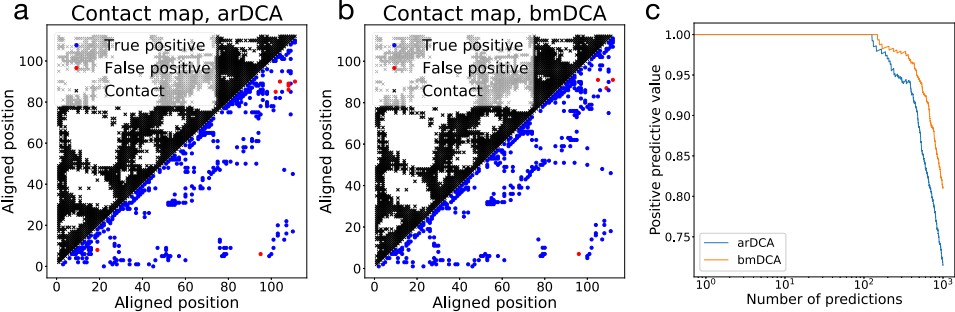

**Fig. 4 Prediction of residue-residue contacts by arDCA as compared to bmDCA.** Panels **a** and **b** show the true (black, upper triangle) and predicted (lower triangle) contact maps for PF00072, with blue (red) dots indicating true (false) positive predictions. Panel **c** shows the positive predictive values (PPV, fraction of true positives in the first predictions) as a function of the number of predictions.

arDCA might require a computationally heavy averaging over a large ensemble of background sequences. Fortunately, as we have checked, the predicted contacts hardly depend on the reference sequence chosen. It is therefore possible to take any arbitrary reference sequence belonging to the homologous family and determine epistatic couplings relative to this single sequence. This observation causes an enormous speedup by a factor $M$, with $M$ being the depths of the MSA of natural homologs.

The aim of this section was to compare the performance of arDCA in contact prediction when compared to established methods using exactly the same data, i.e., a single MSA of the considered protein family. We have chosen bmDCA in coherence to the rest of the paper, but apart from little quantitative differences, the conclusions remain unchanged when looking to DCA variants based on mean-field or pseudo-likelihood approximations, cf. Supplementary Fig. 9. The recent success of Deep-Learning–based contact prediction has shown that the performance can be substantially improved if coevolution-based contact prediction for thousands of families is combined with supervised learning based on known protein structures, as done by popular methods like RaptorX, DeepMetaPSICOV, AlphaFold, or trRosetta[16–19]. We expect that the performance of arDCA could equally be boosted by supervised learning, but this goes clearly beyond the scope of our work, which concentrates on generative modeling.

**Estimating the size of a family's sequence space.** The MSA of natural sequences contains only a tiny fraction of all sequences, which would have the functional properties characterizing a protein family under consideration, i.e., which might be found in newly sequenced species or be reached by natural evolution. Estimating this number $\mathcal{N}$ of possible sequences, or their entropy $S = \log \mathcal{N}$, is quite complicated in the context of DCA-type pairwise Potts models. It requires advanced sampling techniques[60,61].

In arDCA, we can explicitly calculate the sequence probability $P(a_1, ..., a_L)$. We can therefore estimate the entropy of the corresponding protein family via

$$S = -\sum_{a_1,...,a_L} P(a_1, ..., a_L) \log P(a_1, ..., a_L)$$
$$= \langle E(a_1, ..., a_L) \rangle_P,  \qquad (7)$$

where the second line uses Eq. (4). The ensemble average $\langle \cdot \rangle_P$ can be estimated via the empirical average over a large sequence sample drawn from $P$. As discussed before, extracting i.i.d. samples from arDCA is particularly simple due to their particular factorized form.

Results for the protein families studied here are given in Table 1. As an example, the entropy density equals $S/L = 1.4$ for PF00072. This corresponds to $\mathcal{N} \sim 1.25 \cdot 10^{68}$ sequences. While

being an enormous number, it constitutes only a tiny fraction of all $q^L \sim 1.23 \cdot 10^{148}$ possible sequences of length $L = 112$. Interestingly, the entropies estimated using bmDCA are systematically higher than those of arDCA. On the one hand, this is no surprise: both reproduce accurately the empirical one-residue and two-residue statistics, but bmDCA is a maximum entropy model, which maximizes the entropy given these statistics[4]. On the other hand, our observation implies that the effective multi-site couplings in $E(a_1, .., a_L)$ resulting from the local partition functions $z_i(a_{i-1}, ..., a_1)$ lead to a non-trivial entropy reduction.

## Discussion

We have presented a class of simple autoregressive models, which provide highly accurate and computationally very efficient generative models for protein-sequence families. While being of comparable or even superior performance to bmDCA across a number of tests including the sequence statistics, the sequence distribution in dimensionally reduced principal-component space, the prediction of mutational effects, and residue-residue contacts, arDCA is computationally much more efficient than bmDCA. The particular factorized form of autoregressive models allows for exact likelihood maximization.

It allows also for the calculation of exact sequence probabilities (instead of sequence weights for Potts models). This fact is of great potential interest in homology detection using coevolutionary models, which requires comparing probabilities of the same sequence in distinct models corresponding to distinct protein families. To illustrate this idea in a simple, but the instructive case, we have identified two subfamilies of the PF00072 protein family of response regulators. The first subfamily is characterized by the existence of a DNA-binding domain of the Trans_reg_C protein family (PF00486), the second is by a DNA-binding domain of the GerE protein family (PF00196). For each of the two subfamilies, we have extracted randomly 6000 sequences used to train sub-family specific profile and arDCA models, with $P_1$ being the model for the Trans_reg_C and $P_2$ for the GerE subfamily. Using the log-odds ratio $\log\{P_1(\text{seq})/P_2(\text{seq})\}$ to score all remaining sequences from the two subfamilies, the profile model was able to assign 98.6% of all sequences to the correct subfamily, and 1.4% to the wrong one. arDCA has improved this to 99.7% of correct, and only 0.3% of incorrect assignments, reducing the gray-zone in sub-family assignment by a factor of 3–4. Furthermore, some of the false assignments of the profile model had quite large scores, cf. the histograms in Supplementary Fig. 11, while the false annotations of the arDCA model had scores closer to zero. Therefore, if we consider that a prediction is reliable only if there are no wrong predictions for a larger log-odds ratio score, then the score of arDCA is 97.5% while one of the profile models is only 63.7%.

The importance of accurate generative models becomes also visible via our results on the size of sequence space (or sequence entropy). For the response regulators used as an example throughout the paper (and similar observations are true for all other protein families we analyzed), we find that "only" about $10^{68}$ out of all possible $10^{148}$ amino-acid sequences of the desired length are compatible with the arDCA model, and thus suspected to have the same functionality and the same 3D structure of the proteins collected in the Pfam MSA. This means that a random amino-acid sequence has a probability of about $10^{-80}$ to be actually a valid response-regulator sequence. This number is literally astronomically small, corresponding to the probability of hitting one particular atom when selecting randomly in between all atoms in our universe. The importance of good coevolutionary modeling becomes even more evident when considering all proteins being compatible with the amino-acid conservation patterns in the MSA: the corresponding profile model still results in an effective sequence number of $10^{94}$, i.e., a factor of $10^{26}$ larger than the sequence space respecting also coevolutionary constraints. As was verified in experiments, conservation provides insufficient information for generating functional proteins, while taking coevolution into account leads to finite success probabilities.

Reproducing the statistical features of natural sequences does not necessarily guarantee the sampled sequences to be fully functional protein sequences. To enhance our confidence in these sequences, we have performed two tests.

First, we have reanalyzed the bmDCA-generated sequences of ref. [20], which were experimentally tested for their in-vivo chorismate-mutase activity. Starting from the same MSA of natural sequences, we have trained an arDCA model and calculated the statistical energies of all non-natural and experimentally tested sequences. As is shown in Supplementary Fig. 12, the statistical energies have a Pearson correlation of 97% with the bmDCA energies reported in ref. [20]. In both cases, functional sequences are restricted to the region of low statistical energies.

Furthermore, we have used small samples of 10 artificial or natural response-regulator sequences as inputs for trRosetta[19], in a setting that allows for protein-structure prediction based only on the user-provided MSA, i.e., no homologous sequences are added by trRosetta, and no structural templates are used. As is shown in Supplementary Fig. 13, the predicted structures are very similar to each other, and within a root-mean-square deviation of less than 2 Å from an exemplary PDB structure. The contacts maps extracted from the trRosetta predictions are close to identical.

While these observations do not prove that arDCA-generated sequences are functional or fold into the correct tertiary structure, they are coherent with this conjecture.

Autoregressive models can be easily extended by adding hidden layers in the ansatz for the conditional probabilities $P(a_i|a_{i-1},..., a_1)$, with the aim to increase the expressive power of the overall model. For the families explored here, we found that the one-layer model Eq. (2) is already so accurate, that adding more layers only results in similar, but not superior performance, cf. Supplementary Note 6. However, in longer or more complicated protein families, the larger expressive power of deeper autoregressive models could be helpful. Ultimately, the generative performance of such extended models should be assessed by testing the functionality of the generated sequences in experiments similar to ref. [20].

## Methods

**Inference of the parameters**. We first describe the inference of the parameters via likelihood maximization. In a Bayesian setting, with a uniform prior (we discuss regularization below), the optimal parameters are those that maximize the probability of the data, given as an MSA $\mathcal{D} = (a_i^m | i = 1, ..., L; m = 1, ..., M)$ of $M$

sequences of aligned length $L$:

$$
\begin{aligned}
\{\mathbf{J}^*, \mathbf{h}^*\} &= \arg\max_{\{\mathbf{J}, \mathbf{h}\}} P(\mathcal{D}|\{\mathbf{J}, \mathbf{h}\}) \\
&= \arg\max_{\{\mathbf{J}, \mathbf{h}\}} \log P(\mathcal{D}|\{\mathbf{J}, \mathbf{h}\}) \\
&= \arg\max_{\{\mathbf{J}, \mathbf{h}\}} \sum_{m=1}^{M} \log \prod_{i=1}^{L} P(a_i^m | a_{i-1}^m, ..., a_1^m) \\
&= \arg\max_{\{\mathbf{J}, \mathbf{h}\}} \sum_{m=1}^{M} \sum_{i=1}^{L} \log P(a_i^m | a_{i-1}^m, ..., a_1^m).
\end{aligned}
\tag{8}
$$

Each parameter $h_i(a)$ or $J_{ij}(a,b)$ appears in only one conditional probability $P(a_i|a_{i-1},..., a_1)$, and we can thus maximize independently each conditional probability in Eq. (8):

$$
\begin{aligned}
\{\mathbf{J}_{ij}^*, h_i^*\} &= \arg\max_{\{J_{ij}, h_i\}} \sum_{m=1}^{M} \log P(a_i^m | a_{i-1}^m, ..., a_1^m) \\
&= \arg\max_{\{J_{ij}, h_i\}} \sum_{m=1}^{M} \left[ h_i(a_i^m) + \sum_{j=1}^{i-1} J_{ij}(a_i^m, a_j^m) \right. \\
&\qquad\qquad \left. - \log z_i(a_{i-1}^m, ... a_1^m) \right]
\end{aligned}
$$

where

$$
z_i(a_{i-1}, ... a_1) = \sum_{a_i} \exp\left\{ h_i(a_i) + \sum_{j=1}^{i-1} J_{ij}(a_i, a_j) \right\}
\tag{9}
$$

is the normalization factor of the conditional probability of variable $a_i$.

Differentiating with respect to $h_i(a)$ or to $J_{ij}(a, b)$, with $j = 1, ..., i - 1$, we get the set of equations:

$$
\begin{aligned}
0 &= \frac{1}{M} \sum_{m=1}^{M} \left[ \delta_{a,a_i^m} - \frac{\partial \log z_i(a_{i-1}^m, ... a_1^m)}{\partial h_i(a)} \right], \\
0 &= \frac{1}{M} \sum_{m=1}^{M} \left[ \delta_{a,a_i^m} \delta_{b,a_j^m} - \frac{\partial \log z_i(a_{i-1}^m, ... a_1^m)}{\partial J_{ij}(a, b)} \right],
\end{aligned}
\tag{10}
$$

where $\delta_{a,b}$ is the Kronecker symbol. Using Eq. (9) we find

$$
\begin{aligned}
\frac{\partial \log z_i(a_{i-1}^m, ... a_1^m)}{\partial h_i(a)} &= P(a_i = a | a_{i-1}^m, ..., a_1^m), \\
\frac{\partial \log z_i(a_{i-1}^m, ... a_1^m)}{\partial J_{ij}(a, b)} &= P(a_i = a | a_{i-1}^m, ..., a_1^m) \delta_{a_j^m, b}.
\end{aligned}
\tag{11}
$$

The set of equations thus reduces to a very simple form:

$$
\begin{aligned}
f_i(a) &= \left\langle P(a_i = a | a_{i-1}^m, ..., a_1^m) \right\rangle_{\mathcal{D}}, \\
f_{ij}(a, b) &= \left\langle P(a_i = a | a_{i-1}^m, ..., a_1^m) \delta_{a_j^m, b} \right\rangle_{\mathcal{D}},
\end{aligned}
\tag{12}
$$

where $\langle \bullet \rangle_{\mathcal{D}} = \frac{1}{M} \sum_{m=1}^{M} \bullet^m$ denotes the empirical data average and $f_i(a)$, $f_{ij}(a, b)$ is the empirical one-point and two-point amino-acid frequencies. Note that for the first variable ($i = 1$), which is unconditioned, there is no equation for the couplings, and the equation for the field takes the simple form $f_1(a) = P(a_1 = a)$, which is solved by $h_1(a) = \log f_1(a) + \text{const.}$

Unlike the corresponding equations for the Boltzmann learning of a Potts model[5], there is a mix between probabilities and empirical averages in Eq. (12), and there is no explicit equality between one-point and two-point marginals and empirical one and two-point frequencies. This means that the ability to reproduce the empirical one-point and two-point frequencies are already a statistical test for the generative properties of the model, and not only for the fitting quality of the current parameter values.

The inference can be done very easily with any algorithm using gradient descent, which updates the fields and couplings proportionally to the difference of the two sides of Eq. (12). We used the Low Storage BFGS method to do the inference. We also add an $L2$ regularization, with a regularization strength of $\lambda_J = 10^{-4}$, $\lambda_h = 10^{-6}$ for the generative tests and $\lambda_J = 10^{-2}$, $\lambda_h = 10^{-4}$ for mutational effects and contact prediction. A small regularization leads to better results on generative tests, but a larger regularization is needed for contact prediction of mutational effects. Contact prediction can indeed suffer from too large parameters, and therefore a larger regularization was chosen, coherently with the one used in plmDCA. Note that the gradients are computed exactly at each iteration, as an explicit average over the data, and hence without the need of MCMC sampling. This provides an important advantage over Boltzmann-machine learning.

Finally, in order to partially compensate for the phylogenetic structure of the MSA, which induces correlations among sequences, each sequence is reweighted by a coefficient $w_m$[4]:

$$
\{\mathbf{J}_{ij}^*, h_i^*\} = \arg\max_{\{J_{ij}, h_i\}} \frac{1}{M_{\text{eff}}} \sum_{m=1}^{M} w_m \log P(\mathbf{a}^m | \{J_{ij}, h_i\}),
\tag{13}
$$

which leads to the same equations as above with the only modification of the empirical average as $\langle \bullet \rangle_{\text{data}} = \frac{1}{M_{\text{eff}}} \sum_{m=1}^{M} w_m \bullet^m$. Typically, $w_m$ is given by the

inverse of the number of sequences having least 80% sequence identity with sequence $m$, and $M_{eff} = \sum_m w_m$ denotes the effective number of independent sequences. The goal is to remove the influence of very closely related sequences. Note however that such reweighting cannot fully capture the hierarchical structure of phylogenetic relations between proteins.

**Sampling from the model.** Once the model parameters are inferred, a sequence can be iteratively generated by the following procedure:

- Sample the first residue from $P(a_1)$
- Sample the second residue from $P(a_2|a_1)$ where $a_1$ is sampled in the previous step.

... $L$. Sample the last residue from $P(a_L|a_{L-1}, a_{L-2}, ..., a_2, a_1)$ Each step is very fast because there are only 21 possible values for each probability. Both training and sampling are therefore extremely simple and computationally efficient in arDCA.

**Reporting summary.** Further information on research design is available in the Nature Research Reporting Summary linked to this article.

## Code availability

Codes in Python and Julia are available at https://github.com/pagnani/ArDCA.git.

## Data availability

Data is available at https://github.com/pagnani/ArDCADataand was elaborated using source data freely downloadable from the Pfam database (http://pfam.xfam.org/)[2], cf. Supplementary Table 1. The repository contains also sample MSA generated by arDCA. The input data for Figure 3 are provided by the GEMME paper[58], cf. also Supplementary Table 2.

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

## Acknowledgements

We thank Indaco Biazzo, Matteo Bisardi, Elodie Laine, Anna-Paola Muntoni, Edoardo Sarti, and Kai Shimagaki for helpful discussions and assistance with the data. We especially thank Francisco McGee and Vincenzo Carnevale for providing generated samples from DeepSequence as in ref. [34]. Our work was partially funded by the EU H2020 Research and Innovation Programme MSCA-RISE-2016 under Grant Agreement No. 734439 InferNet (M.W.), and by a grant from the Simons Foundation (#454955, F.Z.). J.T. is supported by a Ph.D. Fellowship of the i-Bio Initiative from the Idex Sorbonne University Alliance.

## Author contributions

A.P., F.Z., and M.W. designed research; J.T., G.U., and A.P. performed research; J.T., G.U., A.P., F.Z., and M.W. analyzed the data; J.T., F.Z., and M.W. wrote the paper.

## Competing interests

The authors declare no competing interests.
