## [Peer Review File · Nature Communications]

Efficient generative modeling of protein sequences using simple autoregressive modelsREVIEWER COMMENTS

Reviewer #1 (Experience: Biochemist, Experimental user's perspective):

The authors argue that generative models are quickly emerging as promising candidates for novel sequence-data driven approaches to protein design, and to obtain structural and functional information about proteins. In this manuscript, the authors propose a simple model architecture called arDCA, that is based on a shallow (one-layer) autoregressive model paired with generalized logistic regression. Such models are computationally efficient and can be developed quickly and reliably. I am enthusiastic about the present manuscript which clearly constitutes an important leap ahead in the field.

My only main criticism is about readability. The manuscript reads as a work meant for a highly specialised audience who is already completely proficient with the problem. The authors have instead sent the manuscript to a journal with a highly diversified readership who will not be able to appreciate 80% of the work presented. While of course the authors should not compromise with quality or rigor, they should do some attempts to explain the math presented and to explain the acronyms (e.g. bmDCA, plmDCA, etc.) that are introduced but not defined.

Reviewer #2 (Experience: Machine/Deep learning to predict protein function from sequence/generative modeling):

Summary:

Trinquier et al. develop an auto-regressive machine learning model for protein families which is based on softmax-regression of both single-residue features (resembling DCA potentials) and pairwise-residue features (resembling DCA couplings). This unsupervised generative model is trained on the MSA of a protein family of interest, with the goal of recapitulating evolutionary conserved and coevolved statistical properties of the family's sequence distribution. The authors show that (1) their model is computationally significantly faster to train than the corresponding DCA-based models, (2) samples drawn from the model recapitulates amino acid frequencies and pairwise amino acid correlations found in the original sequence data, (3) their model marginally outperforms another DCA-based model in mutation effect prediction (but is outperformed by a deep variational autoencoder) and (4) their model is comparable to another unsupervised DCA-based model for tertiary residue-residue contact predictions.

Significance:

Overall, I found this to be a well executed study with a rigorous mathematical and computational methodology to stand on. The most noteworthy result is the significantly reduced training time needed to obtain a model which, for most intents and purposes (as shown empirically on the various tasks), resemble a DCA model and recapitulate the same statistical single- and pairwise amino acid properties. This more efficient and easier-to-train model will likely be of use to researchers in the field who would otherwise use DCA-based modeling.

However, considering each respective task in isolation (mutational effect prediction, residue-residue contact prediction and sequence design), this contribution is probably of limited significance due to the recent advances made by deep learning models, and the observed/presumed superior performance of those approaches.

First, for mutational effect prediction, the authors themselves show that a deep variational autoencoder (DeepSequence) [1], which is also trained in an unsupervised manner on unlabeled data, outperforms their proposed arDCA model. If that's the case, then I do not see the utility in using their model. Granted, the training time is significantly longer for DeepSequence, but I don't think that is the most limiting factor for users, and it is very dependent on machine specifications. Second, for residue-residue contact prediction, the authors make no comparison to the deep learning-based AlphaFold or trRosetta models [2, 3], which are considered state-of-the-art for contact prediction. On the one hand, AlphaFold and trRosetta are trained in a supervised manner on solved structures whereas the authors train their model on unlabeled datasets, so the comparison may not be entirely fair. However, the authors argue in the introduction that training an unsupervised model is preferable since it enables much larger unlabeled datasets to be used. But if AlphaFold and trRosetta outperform this unsupervised model (which I think they will), then there is no added benefit (or much utility) of the authors' new model for contact prediction. Third, in the context of sequence design, the authors state in the introduction that unsupervised deep generative models are largely unexplored for sequence design. However, there are now many approaches based on deep generative models that demonstrate de novo design of protein sequences [4, 5, 6, 7, 8, 9], some of which combine unsupervised training with supervised fine-tuning [6]. The authors make no comparison to these models and it is hard to think that they would perform comparably with this much simpler model.

[1] Riesselman, A.J., Ingraham, J.B. and Marks, D.S., 2018. Deep generative models of genetic variation capture the effects of mutations. *Nature methods*, 15(10), pp.816-822.

[2] Senior, A.W., Evans, R., Jumper, J., Kirkpatrick, J., Sifre, L., Green, T., Qin, C., Žídek, A., Nelson, A.W., Bridgland, A. and Penedones, H., 2020. Improved protein structure prediction using potentials from deep learning. *Nature*, 577(7792), pp.706-710.

[3] Yang, Jianyi, et al. "Improved protein structure prediction using predicted interresidue orientations." *Proceedings of the National Academy of Sciences* 117.3 (2020): 1496-1503.

[4] Riesselman, Adam J., et al. "Accelerating Protein Design Using Autoregressive Generative Models." bioRxiv (2019): 757252.

[5] Greener, Joe G., Lewis Moffat, and David T. Jones. "Design of metalloproteins and novel protein folds using variational autoencoders." Scientific reports 8.1 (2018): 1-12.

[6] Costello, Zak, and Hector Garcia Martin. "How to hallucinate functional proteins." arXiv preprint arXiv:1903.00458 (2019).

[7] Repecka, D., Jauniskis, V., Karpus, L., Rembeza, E., Rokaitis, I., Zrimec, J., Poviloniene, S., Laurynenas, A., Viknander, S., Abuajwa, W. and Savolainen, O., 2021. Expanding functional protein sequence spaces using generative adversarial networks. Nature Machine Intelligence, pp.1-10.

[8] Amimeur, Tileli, et al. "Designing Feature-Controlled Humanoid Antibody Discovery Libraries Using Generative Adversarial Networks." bioRxiv (2020).

[9] Strokach, Alexey, et al. "Fast and flexible protein design using deep graph neural networks." Cell Systems 11.4 (2020): 402-411.

References:

The authors cite previous and related research (including [1, 2, 3, 4] from above). I would suggest adding references to [5, 6, 7, 8, 9] as they are highly relevant for protein sequence design using generative models.

Methodology:

The computational methodology is presented well. The mathematical framework, problem definition, model and optimization are all presented in enough detail to be reproduced.

To strengthen the conclusions and demonstrated applications, I would suggest complementing some of the analyses with additional comparisons. See Suggested Improvements below.

Concerns and Suggested improvements:

Major 1: The analysis in Table 1 and Figure 2 compares the proposed model (arDCA) to another DCA model (bmDCA) on the basis of statistical properties (amino acid frequency, pairwise and triplet

correlations) captured from the sequence data. While this shows that arDCA compares well to bmDCA, the authors should also include comparisons to unsupervised deep generative models such as DeepSequence [1], the auto-regressive model from [4] or the VAE from [6]. This is needed to strengthen the claim that their model performs comparably to deep models.

Major 2: From the current analysis in Figure 3, it seems that the deep generative model DeepSequence [1] moderately outperforms the proposed model on average, although for certain families the improvement in correlation is significantly higher. It would be good to complement this with a more in-depth analysis of exactly what type of mutations the deeper model seems to perform better at. Does arDCA systematically miss specific types of statistical relationships in the data that DeepSequence can learn?

Major 3: For the residue-residue contact prediction task (Figure 4), I suggest that the authors include comparisons to AlphaFold [2] or trRosetta [3], in order to estimate to what extent (if any) such deep (supervised) models outperform the authors' unsupervised DCA-based model.

Minor 1: The authors make a loose claim on page 5 that "The four context-dependent models perform in a very similar way; their predictions are usually more correlated to each other than to the experimental data". Do the authors mean the models overfit to the training data? I suggest either removing or clarifying this claim with supporting data.

Minor 2: In addition to the amino acid statistics reported in Table 1 and Figure 2, I would suggest performing additional comparisons between original sequences from the data and samples from the proposed model based on biophysical properties. For example, comparing the distributions of hydrophobicity indices, instability indices or secondary structure features. These additional comparisons would raise credibility in that the proposed model has captured relevant biophysical properties from the data.

Minor 3: In both the introduction and in the discussion, the authors claim that being able to efficiently compute the normalizer z (which allows them to estimate normalized probabilities) is of great potential importance for homology detection using coevolutionary models, but the authors never present any analysis demonstrating that this would work well. So I suggest removing or rephrasing the claim "..., thereby enabling the comparison of the same sequence in different models for different protein families." at the end of the introduction.

Reviewer #3 (Expertise: Machine/Deep learning to predict protein function from sequence/generative modeling):

The work by Trinquier et al. deals with the development of generative models of protein sequence families using supervised learning methodologies. This study advances the field of amino acid coevolution pioneered by some of the authors and provides an original solution to reduce the computational burden that the state of the art sequence models are characterized with. The usefulness of sequence Potts models like Direct Coupling Analysis has been evident in recent years to devise protein structures, interactions and to approximate fitness landscapes for biomolecules. As the field shifts towards protein design, generative methodologies that agree better with the global site statistics in a family are needed. The promise is that such models will be more accurate identifying functional sequences, and the authors provide evidence of these in their previous work combining generative models like bmDCA with experimental protein design. The most important contribution of this article is the fact that such generative models are not only expensive but also in many cases do not converge, therefore a novel methodology that could resolve this challenge is a very important contribution to the field.

In my opinion, the methodological advances proposed in the “autoregressive DCA” or arDCA are not only novel but will also accelerate the research in protein landscapes and design in a meaningful way for the scientific community, maybe in a parallel way on how DCA accelerated the field of residue coevolution. Although, I find these results very important and of note, I felt that the authors left behind important analyses that could help stress the main contribution of their work, i.e. the reduction in computational complexity and convergence. The authors focused mostly in showcasing how arDCA is as good as bmDCA, which is important to show but they did not formally show how arDCA provides advantages over bmDCA or other generative models. I don’t see this as an important drawback but simply an issue that can be resolved in a revised version of this manuscript. I also have a series of general and minor comments that I would like to be addressed in a new version of this article.

General Comments

1. I suggest removing non-objective wording like “extremely”, “astonishingly”, “excellent” and replace with more descriptive adjectives
2. The abstract mentions a “substantially lower computational cost” , please try to add a more quantitative statement, maybe with some general improvement numbers

3. In the discussion of the effect of mutations reference [14] is cited, in my opinion a better or additional reference showcasing this concept would be Cheng et al. *Molecular Biology and Evolution*. 2019 (doi.org/10.1093/molbev/msw188)

4. The possibility of calculating exact sequence probabilities is quite interesting and a unique contribution of this work. Could the authors provide more details on how its use in sequence homology can benefit from this feature of arDCA? Maybe provide an example where using non-exact probabilities provides misleading or inconclusive information? This would certainly highlight in a stronger way the unique contributions of this methodology.

5. When the authors mention the application of generative models, particularly bmDCA, they might also suggest its recent use in models of evolution as presented in De la Paz et al. *PNAS*. 2020 (doi.org/10.1073/pnas.1913071117)

5. The authors mention that for the case of pseudo likelihood maximization DCA (plmDCA) having parameter symmetrization causes accuracy loss. Could the authors provide more details on this statement? Maybe an example that can illustrate this drawback in the plmDCA formulation?

6. The authors make their case when comparing arDCA versus bmDCA which yields accurate statistics. However, they focus primarily on well behaved cases like the response regulator. It would be very interesting to compare the behavior of arDCA/bmDCA for cases like the Leptin family or the ObR_IG (Obesity receptor) where the number of available sequences is more limited and bmDCA has shown some convergence problems. An analysis on these cases could strengthen the premise of the utility of arDCA.

7. Following up on the previous comments, the authors should also include a computational complexity analysis for some (not necessarily all) of the families described in the paper to showcase the advantages in running time for bmDCA (some is already covered in the SI). Using the families in comment 6 could also help compare questions about convergence. It would also be useful to include a comparison with another generative model called Mi3 from Haldane and Levy (doi.org/10.1016/j.cpc.2020.107312) which uses GPUs to accelerate the sampling process.

8. The manuscript mentions that the generative process of arDCA is more or less equivalent to bmDCA. Given that bmDCA was used to generate functional sequences of the chromate mutase enzymes. The authors could compare if some of those functional sequences also score well in the probability distribution of arDCA, this could give some evidence that arDCA could in fact produce functional proteins.

9. An important claim is that the energy of equation (4) is not a simple sum of parameters but also includes the logarithms of the local partition functions. Could the authors give more details on how, in practical terms, this is better or more useful. Is there an example that can show the relevance of one approach versus the other?

10. The performance on mutational effects is really encouraging especially for viral proteins. Could the authors provide a rationale of why this method could be better compared to others when dealing with less diverse sequences like the viral ensemble?

11. Fig. 4 compares contact prediction of arDCA vs. bmDCA. Could the authors also include a comparison with respect to mfDCA and plmDCA? That way we would be able to compare arDCA against other non-generative methods.

12. To me, one of the most interesting contributions of this work is the ability to compute the entropy density of a family efficiently and maybe more accurately. I suggest to include this contribution in the abstract.

13. Given that the advantage of family comparison due to exact probabilities is mentioned several times, I suggest including an example that could help showcase this in a more concrete way.

14. Figure 1 in the SI seems to me quite important, I suggest to bring this back to the main text as well with further analysis on computational complexity of the methods.

15. In several sections a “profile” model is mentioned, could you give more details on this? Is this based on Hidden Markov Models?

16. Make sure to mention in the main text that there is no real advantage for the two-layer model in generative qualities or computational time.

Minor Comments

1. Introduction. Pfam 34 now has about 19,179 families, please update the number in the introduction.
2. Results are showcased for the Family PF00076, please explicitly state which family of proteins is this.
3. References to the SI should be more specific, e.g. include figure numbers, section names/numbers to facilitate the reference to the reader.
4. For the section name in the SI , I suggest changing the title to "Parameter Inference"
5. In section I of the SI. What was the criterion to select the L2 regularization strengths ?
6. In the SI, when describing algorithms, I suggest numbering the steps instead of using bullets
7. In the SI section IV. A-4, subheading a and b, please add a period after PPV and Contact map
8. In SI section "Families used for mutational effects", a "standard laptop", is mentioned. It would be good to include the specification of such laptop given that the standard laptop specifications change every 6 months.
9. In SI section V. Positional Order. Is the "natural order" the same as "direct order"? please clarify.
10. In SI section VI. Replace "Two layer autoregressive models" with "Two-layer autoregressive models"
11. In SI, the first sentence after equation 9, replace "is are parameter matrices" with "are parameter matrices"

We thank all reviewers for their careful reading of our manuscript, and for the suggestions, which have helped to improve contents and presentation. We have taken into account all points mentioned, cf. the detailed point-by-point responses below - the original reports are reproduced in black, our answers given in blue. We have also provided a pdf of our manuscript with highlighted changes compared to our original submission.

Reviewer #1 (Experience: Biochemist, Experimental user's perspective):

The authors argue that generative models are quickly emerging as promising candidates for novel sequence-data driven approaches to protein design, and to obtain structural and functional information about proteins. In this manuscript, the authors propose a simple model architecture called arDCA, that is based on a shallow (one-layer) autoregressive model paired with generalized logistic regression. Such models are computationally efficient and can be developed quickly and reliably. I am enthusiastic about the present manuscript which clearly constitutes an important leap ahead in the field.

My only main criticism is about readability. The manuscript reads as a work meant for a highly specialised audience who is already completely proficient with the problem. The authors have instead sent the manuscript to a journal with a highly diversified readership who will not be able to appreciate 80% of the work presented. While of course the authors should not compromise with quality or rigor, they should do some attempts to explain the math presented and to explain the acronyms (e.g. bmDCA, plmDCA, etc.) that are introduced but not defined.

We thank the reviewer for the positive evaluation of our manuscript. We have made an effort to make the paper more readable for a broader audience. To reach this, we have extended the introduction by a section about the state of the art, which describes better why we do this study, and why we think that autoregressive models are an interesting new step in the statistical modeling of protein families. We need, however, also to keep the technical clarity of the presentation. What we propose is a method more than a new biological insight, so necessarily the technical aspects have to be discussed. We hope that the relative weights we give now to readability vs. technical detail and exactness satisfy the request of the reviewer.

Reviewer #2 (Experience: Machine/Deep learning to predict protein function from sequence/generative modeling):

Summary:

Trinquier et al. develop an auto-regressive machine learning model for protein families which is based on softmax-regression of both single-residue features (resembling DCA potentials) and pairwise-residue features (resembling DCA couplings). This unsupervised generative model is trained on the MSA of a protein family of interest, with the goal of recapitulating evolutionary conserved and coevolved statistical properties of the family's sequence distribution. The authors show that (1) their model is computationally significantly faster to train than the corresponding DCA-based models, (2) samples drawn from the model recapitulates amino acid frequencies and pairwise amino acid correlations found in the original sequence data, (3) their model

marginally outperforms another DCA-based model in mutation effect prediction (but is outperformed by a deep variational autoencoder) and (4) their model is comparable to another unsupervised DCA-based model for tertiary residue-residue contact predictions.

Significance:

Overall, I found this to be a well executed study with a rigorous mathematical and computational methodology to stand on. The most noteworthy result is the significantly reduced training time needed to obtain a model which, for most intents and purposes (as shown empirically on the various tasks), resemble a DCA model and recapitulate the same statistical single- and pairwise amino acid properties. This more efficient and easier-to-train model will likely be of use to researchers in the field who would otherwise use DCA-based modeling.

However, considering each respective task in isolation (mutational effect prediction, residue-residue contact prediction and sequence design), this contribution is probably of limited significance due to the recent advances made by deep learning models, and the observed/presumed superior performance of those approaches.

First, for mutational effect prediction, the authors themselves show that a deep variational autoencoder (DeepSequence) [1], which is also trained in an unsupervised manner on unlabeled data, outperforms their proposed arDCA model. If that's the case, then I do not see the utility in using their model. Granted, the training time is significantly longer for DeepSequence, but I don't think that is the most limiting factor for users, and it is very dependent on machine specifications. Second, for residue-residue contact prediction, the authors make no comparison to the deep learning-based AlphaFold or trRosetta models [2, 3], which are considered state-of-the-art for contact prediction. On the one hand, AlphaFold and trRosetta are trained in a supervised manner on solved structures whereas the authors train their model on unlabeled datasets, so the comparison may not be entirely fair. However, the authors argue in the introduction that training an unsupervised model is preferable since it enables much larger unlabeled datasets to be used. But if AlphaFold and trRosetta outperform this unsupervised model (which I think they will), then there is no added benefit (or much utility) of the authors' new model for contact prediction. Third, in the context of sequence design, the authors state in the introduction that unsupervised deep generative models are largely unexplored for sequence design. However, there are now many approaches based on deep generative models that demonstrate de novo design of protein sequences [4, 5, 6, 7, 8, 9], some of which combine unsupervised training with supervised fine-tuning [6]. The authors make no comparison to these models and it is hard to think that they would perform comparably with this much simpler model.

We thank the reviewer for this overall positive evaluation of our work, and for bringing up the possible limitations. It is clear that our approach is not optimized for any of the specific applications, since it is purely based on sequence information and aims at reproducing the sequence statistics. Extracting biological predictions like mutational effects or contacts are done in the unsupervised way of standard DCA, and should therefore in particular be compared to

this approach. Concerning the points mentioned by the reviewer in the last paragraph, we have added some analyses:

(i) We have added a more detailed comparison with DeepSequence for the mutational-effect prediction. The new Fig.3 shows that DeepSequence and arDCA perform very similarly, possibly with an advantage for DeepSequence for large datasets, and arDCA for smaller datasets. However, the differences remain small, and in all cases the prediction of DeepSequence and arDCA are more correlated to each other than to the experimental measurements (see the new Fig.S7). It should also be noted that, as shown in arXiv:2012.02296, bmDCA (and thus arDCA, because they have comparable generative quality) significantly outperforms DeepSequence in generating homologous sequences with low sequence identity, i.e. in reproducing the overall statistics of the natural sequences. We have been able to confirm these results for the protein families of our study.

(ii) Most of the best deep-learning (DL) based contact predictors (including trRosetta, AlphaFold, RaptorX, DeepMetaPSICOV) first perform unsupervised coevolutionary analysis as done also by the different implementations of DCA, and use the analysis for many protein families, together with PDB structures, for a better supervised extraction of contacts from coevolutionary signals. Our arDCA does only address the first step, and does not use MSA and structures of other families than the currently studied one. So we agree that for contact prediction, trRosetta etc. are always to be preferred over arDCA. However, what we show in the manuscript is that arDCA performs as well as the coevolutionary predictors *using the same input data*. The benefit for contact prediction could be to use arDCA in the data-preparation for DL, but this is definitely a work going beyond the scope of the current paper - which is generative modeling.

In this context, the reviewer's remark has brought us to an interesting and potentially relevant idea for testing the generative properties from a structural perspective: are generated sequences predicted to fold into the same structure as the natural ones? To assess this, we have used trRosetta to predict structures for limited MSA of 10 natural or 10 artificial sequences. We have chosen trRosetta, since the web server allows for user-provided MSA (without enlarging them with natural homologs), and for opting out the use of templates. Interestingly we find that the generated sequences are predicted to fold into almost the same 3D structure, with RMSD below 2Å to example PDB structures from the same family. We added these results in the discussion section, and in a new Fig.S13.

[1] Riesselman, A.J., Ingraham, J.B. and Marks, D.S., 2018. Deep generative models of genetic variation capture the effects of mutations. *Nature methods*, 15(10), pp.816-822.

[2] Senior, A.W., Evans, R., Jumper, J., Kirkpatrick, J., Sifre, L., Green, T., Qin, C., Žídek, A., Nelson, A.W., Bridgland, A. and Penedones, H., 2020. Improved protein structure prediction using potentials from deep learning. *Nature*, 577(7792), pp.706-710.

[3] Yang, Jianyi, et al. "Improved protein structure prediction using predicted interresidue orientations." *Proceedings of the National Academy of Sciences* 117.3 (2020): 1496-1503.

[4] Riesselman, Adam J., et al. "Accelerating Protein Design Using Autoregressive Generative Models." *bioRxiv* (2019): 757252.

[5] Greener, Joe G., Lewis Moffat, and David T. Jones. "Design of metalloproteins and novel protein folds using variational autoencoders." *Scientific reports* 8.1 (2018): 1-12.

- [6] Costello, Zak, and Hector Garcia Martin. "How to hallucinate functional proteins." arXiv preprint arXiv:1903.00458 (2019).
- [7] Repecka, D., Jauniskis, V., Karpus, L., Rembeza, E., Rokaitis, I., Zrimec, J., Poviloniene, S., Laurynenas, A., Viknander, S., Abuajwa, W. and Savolainen, O., 2021. Expanding functional protein sequence spaces using generative adversarial networks. Nature Machine Intelligence, pp.1-10.
- [8] Amimeur, Tileli, et al. "Designing Feature-Controlled Humanoid Antibody Discovery Libraries Using Generative Adversarial Networks." bioRxiv (2020).
- [9] Strokach, Alexey, et al. "Fast and flexible protein design using deep graph neural networks." Cell Systems 11.4 (2020): 402-411.

References:

The authors cite previous and related research (including [1, 2, 3, 4] from above). I would suggest adding references to [5, 6, 7, 8, 9] as they are highly relevant for protein sequence design using generative models.

We have followed the suggestion of the reviewer and cited the aforementioned papers, together with some additional new references. Note that following the comments of Reviewer #1, we have added a new section reviewing the state of the art, where most of these references are now discussed.

Methodology:

The computational methodology is presented well. The mathematical framework, problem definition, model and optimization are all presented in enough detail to be reproduced.

To strengthen the conclusions and demonstrated applications, I would suggest complementing some of the analyses with additional comparisons. See Suggested Improvements below.

Concerns and Suggested improvements:

Major 1: The analysis in Table 1 and Figure 2 compares the proposed model (arDCA) to another DCA model (bmDCA) on the basis of statistical properties (amino acid frequency, pairwise and triplet correlations) captured from the sequence data. While this shows that arDCA compares well to bmDCA, the authors should also include comparisons to unsupervised deep generative models such as DeepSequence [1], the auto-regressive model from [4] or the VAE from [6]. This is needed to strengthen the claim that their model performs comparably to deep models.

We thank the reviewer for these remarks. We have found a recent reference [arXiv:2012.02296], which has compared Mi3, a GPU-based implementation of Boltzmann machine learning (i.e. an approach equivalent to bmDCA) with DeepSequence and their own implementation of VAE. They find that Potts models outperform DeepSequence in their generative capacity, according to various measures. We now cite this observation in the

introduction, and confirm the results also in comparison to arDCA for the protein families used in our study (cf. new results added to Table I). We have also looked into the other generative models suggested by the referee. In particular the deep autoregressive model of [4] would be interesting to compare to our simple autoregressive architecture. Unfortunately the provided code is defective (a problem known to the authors, but currently not corrected), and the code of [6] is not yet publicly available. We thus conclude that the comparison to bmDCA in the generative capacity is the most stringent comparison we were able to perform. We have elaborated this argument in the revised manuscript, see Sec.IIC.

Major 2: From the current analysis in Figure 3, it seems that the deep generative model DeepSequence [1] moderately outperforms the proposed model on average, although for certain families the improvement in correlation is significantly higher. It would be good to complement this with a more in-depth analysis of exactly what type of mutations the deeper model seems to perform better at. Does arDCA systematically miss specific types of statistical relationships in the data that DeepSequence can learn?

We have performed a more quantitative comparison between DeepSequence and arDCA for mutational effect prediction, as already mentioned above. We find that both perform systematically in a very similar way, with a scatter around the diagonal in the new Fig. 3B. We observe a slight advantage for DeepSequence in large MSA, and for arDCA in small MSA, as to be expected from model complexity.

We have tried to pin down the differences between the two methods. We found, however, no striking systematic and interpretable difference. On the contrary, DeepSequence and arDCA are in general highly correlated to each other, with correlations above 80% (cf. new supplementary Fig.S7), i.e. much higher than to most experimental datasets, which measure specific phenotypes providing only rough proxies for fitness. Viral families are exceptions, in which arDCA is much less correlated with DeepSequence (Fig.S7), and for these families arDCA definitely outperforms DeepSequence (Fig.3B).

We conclude that arDCA would be the preferred method when going to large-scale studies (due to its computational efficiency) or to limited datasets, while DeepSequence would be expected to perform better in the case of very deep MSA. We have modified the manuscript to include the new figures comparing the two methods (Fig.3B and S7), and the conclusion on the preferred usage in mutational effect prediction (Sec.IID).

Major 3: For the residue-residue contact prediction task (Figure 4), I suggest that the authors include comparisons to AlphaFold [2] or trRosetta [3], in order to estimate to what extent (if any) such deep (supervised) models outperform the authors' unsupervised DCA-based model.

Since deep-learning based contact predictors (using many MSA and PDBs for training) are proven in numerous studies to outperform unsupervised contact prediction by methods like DCA, Gremlin, PSICOV etc., and arDCA performs just as good as DCA, the conclusion is clear: Do not use arDCA for contact prediction, it is not intended for this application! Our comparison

was thought to compare with an established method *using exactly the same input data* - a single MSA for the family of interest. We have made this idea very clear in our revision (Sec.IIE).

However, as mentioned above, arDCA is a generative model, so the main question is if generated sequences are good sequences. So we have used trRosetta, in a version suitable for benchmarking sequences using only user-provided MSA and no structural templates, and found that the predicted structure for small artificial MSA is almost identical to the one predicted for a small sub-MSA of natural sequences, and below 2Å RMSD to an example PDB of a protein from the same family (PF00072). This is comparable to the variability between different exemplary PDB structures for different proteins of one family. We have added this observation to the discussion section of our paper, and corroborated the findings by new supplementary Fig.S13.

Minor 1: The authors make a loose claim on page 5 that “The four context-dependent models perform in a very similar way; their predictions are usually more correlated to each other than to the experimental data”. Do the authors mean the models overfit to the training data? I suggest either removing or clarifying this claim with supporting data.

As we have mentioned before, we have performed a deeper comparison between DeepSequence and arDCA, now reported in Fig.S7. We would not say that the methods overfit the data, but more optimistically that they extract similar information - which however has only limited correlation to the specific experimental phenotype (frequently protein stability of ligand binding affinity, and not protein “fitness”). We think that this is an intrinsic limitation because when using sequence data, we extract common evolutionary constraints acting on proteins over evolutionary timescales and under different conditions in different species, while DMS experiments measure changes in a specific phenotype in a specific protein under single amino-acid substitutions.

We have clarified this point in our revision. We would need complementary work overcoming this common discrepancy of all MSA-based predictors with DMS experiments, but this question goes clearly beyond the scope of our paper.

Minor 2: In addition to the amino acid statistics reported in Table 1 and Figure 2, I would suggest performing additional comparisons between original sequences from the data and samples from the proposed model based on biophysical properties. For example, comparing the distributions of hydrophobicity indices, instability indices or secondary structure features. These additional comparisons would raise credibility in that the proposed model has captured relevant biophysical properties from the data.

As mentioned above, we have done structure prediction using trRosetta for the generated artificial sequences. The high similarity to sample structures for the studied protein family, and to predicted structures from natural data, implies directly that important biophysical properties (like secondary structure, solvent accessibility, hydrophobic core) are also reproduced. We could add

more direct predictors of these properties if required by the reviewer, but we hope that the predicted structure for the artificial sequences is already a convincing argument.

Minor 3: In both the introduction and in the discussion, the authors claim that being able to efficiently compute the normalizer z (which allows them to estimate normalized probabilities) is of great potential importance for homology detection using coevolutionary models, but the authors never present any analysis demonstrating that this would work well. So I suggest removing or rephrasing the claim "..., thereby enabling the comparison of the same sequence in different models for different protein families." at the end of the introduction.

To support our claim, we have included a simple experiment: We have extracted from the response-regulator (RR) family PF00072 two subfamilies, characterized by the presence of two distinct DNA-binding domains in the same protein. While belonging to the same RR family, the corresponding subfamilies have slightly different properties: while having highly similar protein folds, they form structurally different homodimers. From each subfamily, we have randomly extracted 2000 proteins, and learned sub-family specific profile and arDCA models. In turn, we have computed the log-odds ratios $\log(P(\text{seq}|\text{OmpR}) / P(\text{seq}|\text{GerE}))$ for all sequences from the subfamilies. The results are reported in the new Fig.S11. A profile model performs already very well for this case (3% of false positive subfamily assignments), but the arDCA model almost solves the problem – only 0.3% of the proteins are wrongly assigned. We also observe that the profile model makes some strong errors ($|\log \text{ odds}| \sim 50$), while the errors in arDCA are very close to the decision threshold.

We hope that this simple experiment illustrates the potential, which should be exploited in future work. In the current manuscript, we now discuss this experiment to illustrate the point, in the discussion section and in Fig.S11.

Reviewer #3 (Expertise: Machine/Deep learning to predict protein function from sequence/generative modeling):

The work by Trinquier et al. deals with the development of generative models of protein sequence families using supervised learning methodologies. This study advances the field of amino acid coevolution pioneered by some of the authors and provides an original solution to reduce the computational burden that the state of the art sequence models are characterized with. The usefulness of sequence Potts models like Direct Coupling Analysis has been evident in recent years to devise protein structures, interactions and to approximate fitness landscapes for biomolecules. As the field shifts towards protein design, generative methodologies that agree better with the global site statistics in a family are needed. The promise is that such models will be more accurate identifying functional sequences, and the authors provide evidence of these in their previous work combining generative models like bmDCA with experimental protein design. The most important contribution of this article is the fact that such generative models are not only expensive but also in many cases do not converge, therefore a novel methodology that could resolve this challenge is a very important contribution to the field.

In my opinion, the methodological advances proposed in the “autoregressive DCA” or arDCA are not only novel but will also accelerate the research in protein landscapes and design in a meaningful way for the scientific community, maybe in a parallel way on how DCA accelerated the field of residue coevolution. Although I find these results very important and of note, I felt that the authors left behind important analyses that could help stress the main contribution of their work, i.e. the reduction in computational complexity and convergence. The authors focused mostly on showcasing how arDCA is as good as bmDCA, which is important to show but they did not formally show how arDCA provides advantages over bmDCA or other generative models. I don’t see this as an important drawback but simply an issue that can be resolved in a revised version of this manuscript. I also have a series of general and minor comments that I would like to be addressed in a new version of this article.

We are grateful to the reviewer for the positive evaluation of our work, and for pointing out open questions and current drawbacks in our manuscript. We have carefully revised our manuscript to address all points, cf. the point-by-point reply below.

General Comments

1. I suggest removing non-objective wording like “extremely”, “astonishingly”, “excellent” and replace with more descriptive adjectives

We have removed this kind of wording to reach more objective formulations.

2. The abstract mentions a “substantially lower computational cost” , please try to add a more quantitative statement, maybe with some general improvement numbers

Table 1 contains running times for arDCA vs. bmDCA, with updated numbers using the latest improvements in our implementation. They show that arDCA is about 2-3 orders of magnitude faster than bmDCA when run on a single CPU (we added the specification of the CPU used for the runs). We have now also included in Sec.IIA a discussion of these times and a comparison with Mi3, an efficient implementation of Boltzmann-machine learning using GPUs, which still remains substantially slower than arDCA running on a simple standard CPU.

3. In the discussion of the effect of mutations reference [14] is cited, in my opinion a better or additional reference showcasing this concept would be Cheng et al. Molecular Biology and Evolution. 2019 (doi.org/10.1093/molbev/msw188)

We have added the reference. Given the early publication date of [14] as compared to most other papers in the field, we wanted to keep this reference.

4. The possibility of calculating exact sequence probabilities is quite interesting and a unique contribution of this work. Could the authors provide more details on how its use in sequence homology can benefit from this feature of arDCA? Maybe provide an example where using

non-exact probabilities provides misleading or inconclusive information? This would certainly highlight in a stronger way the unique contributions of this methodology.

To support our claim, we have included a simple experiment: We have extracted from the response-regulator (RR) family PF00072 two subfamilies, characterized by the presence of two distinct DNA-binding domains in the same protein. While belonging to the same RR family, the corresponding subfamilies have slightly different properties: while having highly similar protein folds, they form structurally different homodimers. From each subfamily, we have randomly extracted 2000 proteins, and learned sub-family specific profile and arDCA models. In turn, we have computed the log-odds ratios $\log(P(\text{seq}|\text{OmpR}) / P(\text{seq}|\text{GerE}))$ for all sequences from the subfamilies. The results are reported in the new Fig.S11. A profile model performs already very well for this case (3% of false positive subfamily assignments), but the arDCA model almost solves the problem – only 0.3% of the proteins are wrongly assigned. We also observe that the profile model makes some strong errors ($|\log \text{ odds}| \sim 50$), while the errors in arDCA are very close to the decision threshold.

We hope that this simple experiment illustrates the potential, which should be exploited in future work. In the current manuscript, we now discuss this experiment to illustrate the point, in the discussion section and in Fig.S11.

5. When the authors mention the application of generative models, particularly bmDCA, they might also suggest is recent use in models of evolution as presented in De la Paz et al. PNAS. 2020 (doi.org/10.1073/pnas.1913071117)

We have included the reference in the introduction, together with a remark that generative models can be used as underlying sequence landscapes for data-driven evolutionary models.

5. The authors mention that for the case of pseudo likelihood maximization DCA (plmDCA) having parameter symmetrization causes accuracy loss. Could the authors provide more details on this statement? Maybe an example that can illustrate this drawback in the plmDCA formulation?

plmDCA infers parameters by maximising the pseudo-likelihood. When symmetrizing the parameters, we move parameters away from their PLM values, getting therefore a symmetric but somewhat uncontrolled model. Using only the conditional probability for a_i given the sequence context a_{-i} for predicting mutational effects in site i is actually slightly more accurate than using the standard energy difference in the symmetrized Potts model. While this is an interesting observation on its own, we think it does not fit into the paper, which concentrates on arDCA rather than improving plmDCA-based mutational predictions.

We have reformulated the statement in a more cautious way in the revised manuscript, and added a reference to Ref.[6] where this aspect is discussed in detail.

6. The authors make their case when comparing arDCA versus bmDCA which yields accurate statistics. However, they focus primarily on well behaved cases like the response regulator. It would be very interesting to compare the behavior of arDCA/bmDCA for cases like the Leptin family or the ObR_IG (Obesity receptor) where the number of available sequences is more limited and bmDCA has shown some convergence problems. An analysis on these cases could strengthen the premise of the utility of arDCA.

We performed bmDCA and arDCA training on Leptin and ObR_IG. arDCA does not suffer from convergence problems and it gives reliable generative results, but it does not cure the standard problem of inaccurate contact prediction from small MSA.

For bmDCA, the MCMC sampling becomes non-ergodic during training, which causes very long equilibration times. As a consequence, the resampling from bmDCA gives very poor results: the Pearson is high during training (0.99) but low during resampling (0.43 and 0.62, respectively, for the two families).

So, we can conclude that arDCA outperforms bmDCA in those cases. We added a discussion of this point in the Supplementary Section S3.

7. Following up on the previous comments, the authors should also include a computational complexity analysis for some (not necessarily all) of the families described in the paper to showcase the advantages in running time for bmDCA (some is already covered in the SI). Using the families in comment 6 could also help compare questions about convergence. It would also be useful to include a comparison with another generative model called Mi3 from Haldane and Levy (doi.org/10.1016/j.cpc.2020.107312) which uses GPUs to accelerate the sampling process.

Table 1 contains running times for arDCA vs. bmDCA, with updated numbers using the latest improvements in our implementation. They show that arDCA is about 2-3 orders of magnitude faster than bmDCA when run on a single CPU (we added the specification of the CPU used for the runs). We have now also included in Sec.IIA a discussion of these times and a comparison with Mi3, an efficient implementation of Boltzmann-machine learning using GPUs, which still remains substantially slower than arDCA running on a simple standard CPU.

8. The manuscript mentions that the generative process of arDCA is more or less equivalent to bmDCA. Given that bmDCA was used to generate functional sequences of the chromate mutase enzymes. The authors could compare if some of those functional sequences also score well in the probability distribution of arDCA, this could give some evidence that arDCA could in fact produce functional proteins.

We thank the reviewer for bringing up this point, the results increase the confidence in the arDCA model: For the synthetic chorismate mutases, we find a strong correlation of 97% between the energies of the bmDCA model used in Russ et al. and the arDCA learned on the same natural MSA. This leads to the fact that also for arDCA, high-energy sequences are not

functional, while low-energy sequences may be functional. We have included this observation into the discussion of our paper, and added a new figure S12 to the SI.

9. An important claim is that the energy of equation (4) is not a simple sum of parameters but also includes the logarithms of the local partition functions. Could the authors give more details on how, in practical terms, this is better or more useful. Is there an example that can show the relevance of one approach versus the other?

At this point, it is only a statement of a mathematical fact, which is not necessarily useful or better. The model has the capacity to take higher-order terms into account via these restricted partition functions, but the usefulness of this is hard to quantify.

The real advantage (besides computational efficiency) is in the fact that we can calculate probabilities and not only weights, cf. the answer to Question 4.

10. The performance on mutational effects is really encouraging especially for viral proteins. Could the authors provide a rationale of why this method could be better compared to others when dealing with less diverse sequences like the viral ensemble?

Unfortunately, we do not have a good explanation for this fact, it is more an observation. Following a question of Reviewer #2, we have analyzed the relative performance of Deep Sequence and arDCA. We find them to be similar in performance, with DeepSequence having a little advantage for large, arDCA for small datasets. We think this is an example of the famous bias / variance tradeoff encountered in machine learning, which is in favour of simpler, possibly biased but more robust methods for small datasets, and of models of higher representational power for very large datasets. However, this is more a speculation than an explanation.

The more quantitative comparison of mutational predictions of arDCA and DeepSequence is now part of the manuscript, as new figures 3B and S7.

11. Fig. 4 compares contact prediction of arDCA vs. bmDCA. Could the authors also include a comparison with respect to mfDCA and plmDCA? That way we would be able to compare arDCA against other non-generative methods.

We have included the predictions into the SI, new figure S9, and they confirm our major conclusions up to little quantitative differences. Since these methods have very similar performances, and since our paper concentrates on arDCA as a generative model, we felt that this would overload the figure and the discussion in the main text.

12. To me, one of the most interesting contributions of this work is the ability to compute the entropy density of a family efficiently and maybe more accurately. I suggest to include this contribution in the abstract.

There was information about this in the abstract, but we made the connection between entropy and size of viable sequence space more explicit in the abstract.

13. Given that the advantage of family comparison due to exact probabilities is mentioned several times, I suggest including an example that could help showcase this in a more concrete way.

Cf. our answer to Question 4.

14. Figure 1 in the SI seems to me quite important, I suggest to bring this back to the main text as well with further analysis on computational complexity of the methods.

This figure (now Fig. S4) uses MCMC sampling for the arDCA model, to illustrate its inefficiency as compared to the direct sampling used in the paper, which exploits the positional order in the factorized structure of auto-regressive models. While MCMC is not fully decorrelated even after 10^4 MCMC sweeps, the direct sampling needs to see each position exactly once (i.e. corresponding to a single sweep) to extract a sequence from the model.

Since MCMC sampling is not used in the paper for any arDCA model, we think that inclusion of this figure into the main text would be potentially confusing. However, we have introduced a direct reference to the figure into the main text.

15. In several sections a “profile” model is mentioned, could you give more details on this? Is this based on Hidden Markov Models?

We have now well-defined the term “profile model”, also referred to as “independent-site model” or “fields-only” model.

16. Make sure to mention in the main text that there is no real advantage for the two-layer model in generative qualities or computational time.

We had originally mentioned this observation only in the discussion, now we have added it also to the discussion of the model architecture in Sec. II.A. With our analysis we cannot exclude that some better multi-layer architecture may lead to better results than arDCA, but for sure there has to be some fine-tuning of the precise architecture.

Minor Comments

1. Introduction. Pfam 34 now has about 19,179 families, please update the number in the introduction.

We have updated the numbers.

2. Results are showcased for the Family PF00076, please explicitly state which family of proteins is this.

There was an error in the manuscript, results were actually for the response-regulator family PF00072, results for PF00076 (RRM_1 RNA recognition motif) are part of the SI. We have corrected this error and included the name “response regulator” into the paper at the first appearance of the family.

3. References to the SI should be more specific, e.g. include figure numbers, section names/numbers to facilitate the reference to the reader.

We have made references to the SI more specific.

4. For the section name in the SI, I suggest changing the title to “Parameter Inference”

We have updated the name.

5. In section I of the SI. What was the criterion to select the L2 regularization strengths ?

We used a L2 regularization, with regularization strength of 0.0001 for the generative tests and 0.001 for mutational effects and contact prediction. A small regularization leads to better results on generative tests, but a larger regularization is needed for contact prediction or mutational effects. Contact prediction can indeed suffer from too large parameters, and therefore a larger regularization was chosen, coherently with the one used in PlmDCA. We added this discussion to the SI, Sec.S1.

6. In the SI, when describing algorithms, I suggest numbering the steps instead of using bullets

We have updated the SI accordingly.

7. In the SI section IV. A-4, subheading a and b, please add a period after PPV and Contact map

We have corrected this point.

8. In SI section “Families used for mutational effects”, a “standard laptop”, is mentioned. It would be good to include the specification of such laptop given that the standard laptop specifications change every 6 months.

We have added the specification of the CPU used for the training of the model in Table 1.

9. In SI section V. Positional Order. Is the “natural order” the same as “direct order”? please clarify.

We have replaced “natural order” by “direct order” for full coherence.

10. In SI section VI. Replace “Two layer autoregressive models” with “Two-layer autoregressive models”

We have corrected this point.

11. In SI, the first sentence after equation 9, replace “is are parameter matrices” with “are parameter matrices”

We have corrected this point.

REVIEWERS' COMMENTS

Reviewer #1 (Remarks to the Author):

The manuscript has significantly improved. I recommend it for publication in its present form.

Reviewer #2 (Remarks to the Author):

The authors have made a great effort in updating their manuscript and they have addressed all of my main concerns.

In particular, I applaud them for carrying out additional comparisons to DeepSequence (Table 1 and Fig 3), and for conducting the new homology detection experiment which showcases their model's utility in inferring probabilities. I also believe the new "State-of-the-art" section in the introduction is a great way to introduce the general setting and context to Nature Comms' audience. The trRosetta-folding of arDCA-generated sequences is also very interesting and shows by example the fact that their generative model recapitulates important determinants.

I have no further concerns and recommend accepting the manuscript.

Reviewer #3 (Remarks to the Author):

In this revision, Trinquier et al. address reviewers inquiries and suggestions and produce an updated version of their article on autoregressive models for protein sequence generation. The authors address all my comments and questions in a clear and rigorous manner. I also revised the answers to other reviewers and their responses seem appropriate and in my opinion improved the original article considerably. The new version includes a new section describing the state of the art in the field, something that will be useful for non-specialists but does not compromise the rigor and methodological soundness of the original manuscript. Given this extensive revision with multiple new analyses, figures and the fact that the conclusions were strengthened and the message clarified, I support its publication.